# Telomerase Regulation: A Role for Epigenetics

**DOI:** 10.3390/cancers13061213

**Published:** 2021-03-10

**Authors:** Fatma Dogan, Nicholas R. Forsyth

**Affiliations:** 1The Guy Hilton Research Laboratories, School of Pharmacy and Bioengineering, Faculty of Medicine and Health Sciences, Keele University, Stoke on Trent ST4 7QB, UK; f.dogan@keele.ac.uk; 2School of Medicine, Tongji University, Shanghai 200092, China

**Keywords:** telomerase, TERT, promoter, epigenetics, methylation, cancer

## Abstract

**Simple Summary:**

Maintenance of telomeres is a fundamental step in human carcinogenesis and is primarily regulated by telomerase and the human telomerase reverse transcriptase gene (*TERT*). Improved understanding of the transcriptional control of this gene may provide potential therapeutic targets. Epigenetic modifications are a prominent mechanism to control telomerase activity and regulation of the *TERT* gene. *TERT*-targeting miRNAs have been widely studied and their function explained through pre-clinical in vivo model-based validation studies. Further, histone deacetylase inhibitors are now in pre and early clinical trials with significant clinical success. Importantly, TERT downregulation through epigenetic modifications including *TERT* promoter methylation, histone deacetylase inhibitors, and miRNA activity might contribute to clinical study design. This review provides an overview of the epigenetic mechanisms involved in the regulation of *TERT* expression and telomerase activity.

**Abstract:**

Telomerase was first described by Greider and Blackburn in 1984, a discovery ultimately recognized by the Nobel Prize committee in 2009. The three decades following on from its discovery have been accompanied by an increased understanding of the fundamental mechanisms of telomerase activity, and its role in telomere biology. Telomerase has a clearly defined role in telomere length maintenance and an established influence on DNA replication, differentiation, survival, development, apoptosis, tumorigenesis, and a further role in therapeutic resistance in human stem and cancer cells including those of breast and cervical origin. *TERT* encodes the catalytic subunit and rate-limiting factor for telomerase enzyme activity. The mechanisms of activation or silencing of *TERT* remain open to debate across somatic, cancer, and stem cells. Promoter mutations upstream of *TERT* may promote dysregulated telomerase activation in tumour cells but additional factors including epigenetic, transcriptional and posttranscriptional modifications also have a role to play. Previous systematic analysis indicated methylation and mutation of the *TERT* promoter in 53% and 31%, respectively, of TERT expressing cancer cell lines supporting the concept of a key role for epigenetic alteration associated with TERT dysregulation and cellular transformation. Epigenetic regulators including DNA methylation, histone modification, and non-coding RNAs are now emerging as drivers in the regulation of telomeres and telomerase activity. Epigenetic regulation may be responsible for reversible silencing of TERT in several biological processes including development and differentiation, and increased *TERT* expression in cancers. Understanding the epigenetic mechanisms behind telomerase regulation holds important prospects for cancer treatment, diagnosis and prognosis. This review will focus on the role of epigenetics in telomerase regulation.

## 1. Telomeres and Telomerase Regulation

Telomerase, a ribonucleoprotein enzyme, is the main complex responsible for telomere elongation. The telomeric sequence was recognized as an essential structure at the end of chromosomes [1,2], then sequenced and identified as tandem repeats in the unicellular eukaryote Tetrahymena in 1978 [3]. The telomerase enzyme was discovered in the same model organism [4], followed shortly afterward in human cells [5]. Following on from these pivotal discoveries a substantial research effort has focused on the role of telomerase and telomeres in ageing, cancer and disease. The telomerase complex consists of two main subunits and a range of associated proteins such as Ku, HSP90 and telomerase-associated protein (TP1) [6,7]. The main subunits are the highly conserved, catalytic, human telomerase reverse transcriptase (TERT) protein (1132 amino acids, 127 kDa) [8] and the telomerase RNA component (TERC), where both components are required for telomerase activity [9]. A telomerase-associated protein example, DNA repair protein Ku (a heterodimer of Ku70 and Ku80 subunits), interacts with telomerase through interaction with TERT and TERC subunits [10,11]. TERC component and telomerase-associated proteins are constitutively expressed [12] indicating that enzyme activity depends on transcriptional regulation of *TERT* which is the rate-limiting component of telomerase activity. A direct correlation between *TERT* mRNA level expression and telomerase activity is well described [13].

Genetic mechanisms contribute to the regulation of *TERT* in cancer through promoter mutations [14], gene copy number variations [15], and genomic rearrangements [16]. Epigenetic modifications through promoter methylation [17], histone acetylation and methylation, transcriptional and posttranscriptional mechanisms, and non-coding RNAs [18] all also contribute to the regulation of *TERT* expression in human tumour cell lines [19,20,21].

Telomeres are guanine-rich tandem repeats that protect chromosomes from degradation, provide stabilization, and contribute to overall chromosomal organization [22]. Human telomeres terminate with 3′ single-stranded-DNA overhang which forms a telomere loop stabilized by telomere binding proteins, such as TRF2 (Telomeric repeat-binding factors 2), that protect the telomere end and prevent it from being recognized as a site of DNA damage [23,24]. Telomeric Repeat Binding Factors 1 and 2 (TRF1, TRF2), TRF1-Interacting Nuclear protein 2 (TIN2), repressor activator protein 1 (RAP1), Protection of Telomeres 1 (POT1) and TPP1 telomere protection protein 1 (TPP1) proteins are necessary for telomere function and form the shelterin complex that protects the telomere structure from DNA damage [25]. TRF1, TRF2, and POT1 recognize TTAGGG repeats and the other three scaffold proteins (TIN2, TPP1 and RAP1). Shelterin complex proteins are constitutively expressed, potentially covering all telomeric DNA, and the absence of these proteins can lead to non-homologous end joining (NHEJ), homology-directed repair (HDR) [26], end-to-end fusions [27], genomic instability senescence, or apoptosis [28].

In human cell populations, telomere lengths range from ~5 to 15 kb and each cell division results in a telomere repeat loss of 25–200 bp due to the end replication problem [29,30] In DNA replication, the lagging strand, composed of short DNA fragments and RNA primers, provides a 3’ end for DNA polymerase-driven extension. The end-replication scenario is encountered during the 5`−3` synthesis of DNA because DNA polymerases can only add nucleotides to 3`OH groups. Therefore, the last primer of the lagging strand cannot be synthesized after the RNA primer is removed, resulting in a progressive decrease in telomeric repeats accompanied by cell replication [31]. Proliferative stem cells display regulated telomerase activity and have reduced telomere lengths when compared to germline or hESCs. Cellular senescence was first observed in normal human somatic fibroblasts [32]. Due to the absence of telomerase activity in human somatic cells, there is a progressive loss of telomere repeats, with repeated cell divisions, that ultimately induces an irreversible growth arrest called replicative senescence, or “M1”, when telomere shortening reaches a critical level. However, escape from senescence can occur following the dysregulation of cell cycle checkpoints, such as P53 (TP53), [33], ATM [34] and P16 [35], where progressive proliferation occurs until the cells reach a crisis, or “M2”, which is represented by high levels of apoptosis and genomic instability. Cancer cells can bypass crisis by reactivating telomerase, stabilizing/lengthening telomere ends, and reducing genomic instability levels. They may achieve immortalization via upregulation of telomerase and downregulation of tumour suppressor genes [36]. Telomerase can extend and maintain telomere repeats bypassing the end-replication problem. On the other hand, telomerase activity is not always related to telomere length in cancer cell lines, instead sustaining cell proliferation through telomere length stabilization [37].

Telomerase enzyme activity in somatic cells can be detected at very low levels [38]. However, it is constitutively expressed across germ and stem cells and can be highly expressed in many cancers. To avoid telomere shortening and bypass continued telomeric instability, tumour cells require telomere maintenance mechanisms. Telomerase enzyme activity remains the main mechanism for the maintenance of telomere repeats and 85–90% of cancer cells display telomerase activity [39]. Some immortal human cells maintain telomere length in the absence of telomerase activity with alternative lengthening of telomeres (ALT) mechanism [40]. ALT is based on DNA homologous recombination, and approximately 10–15% of cancers maintain their telomeres in this manner [41]. ALT positive cell lines display highly heterogeneous telomere lengths [42] and contain promyelocytic leukaemia nuclear bodies that comprise extrachromosomal telomeric DNA, proteins associated with DNA recombination and replication processes, and telomere binding proteins [43]. ALT mechanisms are repressed in hESCs but have been suggested to be activated during early development [44]. Telomere length is generally longer in stem cell populations than in somatic and cancer cell lines except for some ALT and telomerase positive cell lines [45].

The *TERT* locus is located on the short arm of human chromosome 5 (5p15.33), 2 Mb distal from the telomere [46]. Longer telomeres have been suggested to fold back on the *TERT* locus thereby negatively regulating expression via telomere position effect [20,47]. Kim and colleagues reported that young human cells with long telomeres have suppressed *TERT* via changes in epigenetic status. Telomere shortening alters histone marks and DNA methylation of promoter regions that regulate *TERT* expression. Human fibroblasts with long telomeres have significantly higher methylation levels in the *TERT* promoter region than cells with shorter telomeres. Further, aged cells with short telomeres show an increase in both active chromatin marks, H3K4 trimethylation (H3K4me3) and H3K9 acetylation (H3K9ac), across the *TERT* promoter. These results support the concept that telomere length-associated changes or telomere position effect might affect *TERT* transcription. Three-dimensional interactions between the *TERT* locus and the sub-telomeric 5p region are conserved in young somatic cells with elongated telomeres but 5p/*TERT* looping interactions become separated by gradual telomere shortening [20]. Telomere looping provides a testable hypothesis to explain how cells turn off enzyme activity during development when telomeres are long and, conversely, reactivation of telomerase in cancer cells that have shorter telomeres [48]. However, human embryonic stem cells (hESCs) have high telomerase activity and long telomeres suggesting that this may be an oversimplification.

## 2. Transcription Factors and Regulation of *TERT* Promoter

Transcriptional regulation of TERT can be driven through epigenetic mechanisms including DNA and histone modifications around the TSS of the promoter. The *TERT* promoter contains a GC-rich sequence and binding sites for many transcription factors but does not contain TATA or CAAT box transcriptional regulatory elements [49,50]. A number of transcription factors (TFs) are known to play roles in the regulation of *TERT* promoter including activators; SP1 (Specificity Protein 1), MYC, NF-κB (Nuclear Factor κB), AP1, STAT3 (Signal Transducer and Activator of Transcription 3), STAT5, PAX (Paired Box Proteins), ER (Estrogen Receptor), HIF1s, and repressors; MAD1, P53, WT1, (Wilms’ tumour 1 suppressor), SP3, CTCF factor and E2F1 [51]. MYC dimerized with MAX has an affinity for the two E-boxes (5′-CACGTG) in the *TERT* promoter [52], while the core promoter has five GC-boxes that are potential SP1 binding sites [53]. Both MYC and SP1 expression correlate with *TERT* transcription in various cancer cell lines [53]. The TERT/SP1 interplay displays further complexity via evidence of co-activation roles on DNMT3B expression in hepatocellular carcinoma (HCC) [54]. However, SP1 plays a dual role via repression of *TERT* promoter in normal human somatic cells [55]. Further, MYC antagonist, MAD1, competitively binds E-boxes and regulates repression of gene expression [56].

STAT3 has a key role in the expression of *TERT* in HCC [57], breast [58] and glioblastoma [59] where STAT3 expression levels correlated with *TERT* expression. STAT3 binding to the promoter in breast cancer stem cells [58] along with STAT5 to the distal promoter region resulted in the activation of TERT expression and telomerase activity [60,61]. NF-κB directly activated *TERT* promoter by binding to the proximal region, and indirectly via increased binding of MYC and SP1 [51,62]. Nuclear hormone receptor ER also played a role in TERT expression and enzyme activity through binding the oestrogen response element in the *TERT* promoter [63,64].

The hypoxia-inducible transcription factor family (HIF1A, HIF2A and HIF3A) are activated in response to sub-ambient oxygen levels and regulate adaptive cellular responses [65]. The *TERT* promoter has two HIF1 consensus sequences between −165 and +51 [66]. HIF1A upregulation induced telomerase activity accompanied by increased *TERT* transcription, driven through these two HIF1 binding motifs [67]. HIF1A and HIF2A significantly increased *TERT* promoter activity in renal cell carcinoma cell lines, while HIF2A alone inhibited *TERT* promoter activity in glioma cell lines [68]. Detailed schematic maps of transcription factors of the *TERT* promoter identify the sites of transcription factors binding, frequent mutation, and response elements that outline a complex landscape of activation, inactivation, and dual role regulators (SP1, HIF2A and EGR1) that can both activate and inactivate gene expression [51].

The P53 transcription factor, with SP1, and potentially others, suppressed *TERT* transcription via binding at −1877 and −1240 upstream of the transcription start site (UTSS). Moreover, P21, itself a downstream activation target of P53, has a significant role in regulating P53 dependent TERT suppression [69,70,71]. Other important transcription factors in the regulation of *TERT* are MYC and MAX which bind the E-boxes consensus sites (5′-CACGTG-3′) at positions −165 and +44 to regulate *TERT* activity [72]. Further, mutations have a significant effect on the transcription of the *TERT* gene. The cytosine to thymidine transitions at −124 bp and −146 bp are common mutations relative to ATG (start codon) [73,74]. These mutations are common across several cancers including melanoma, thyroid cancer, bladder cancer, glioblastoma, HCC and urothelial carcinomas [14,75]. Mutation of the TERT promoter can switch inactive marks to active chromatin marks. The wild-type allele exhibits the H3K27me3 mark associated with epigenetic silencing, while mutant TERT promoters exhibit the H3K4me2/3 mark, which is commonly associated with active chromatin, and enables recruitment of GABPA/B1 [76]. Alternative splicing provides a further level of complexity to the regulation of telomerase, especially during development. Alternative splicing is the process whereby mRNA can be utilized to direct the synthesis of different protein isoforms from a single gene by rearrangement of intron and exon elements. These isoforms can have distinct cellular functions [77,78,79]. Human *TERT* pre-mRNAs can be spliced into 22 isoforms during development but only the full-length transcript (16 exons) is functionally active [79]. A number of isoforms are translated into protein and many have a premature stop codon such as minus beta isoform [79]. The reverse transcriptase domain of TERT contains α, β, or α β alternative spliced isoforms [80]. Expression of splicing variants produces a biologically functional protein containing both α and β regions with a reported correlation between telomerase activity and *TERT* +α+β mRNA levels [81]. Alternative splice variant transcript expression (*TERT* –α+β, *TERT* +α–β and *TERT* –α–β) decreased telomerase activity due to interference from these non-functional alternate splice variants [81]. The –α splicing isoform is translated into a non-functional alternate splice variant without reverse transcriptase activity and its overexpression inhibited telomerase activity [82,83]. Developmentally, elevated telomerase activity correlated with full-length (16 exons) transcript expression in the fetal kidney until gestational week 15. After this time expression of non-functional alternate splice variant (*TERT* –α+β, *TERT* +α–β and *TERT* –α–β) expression becomes dominant and telomerase activity is reduced [81,84,85].

## 3. Epigenetic Control

Waddington defined epigenetics as the correlation between phenotypic and genotypic changes during mammalian embryonic development, and tissue-specific gene function, through studies on Drosophila melanogaster [86,87]. Traditional genetics describe genes and their functions while epigenetics provides a sequence-independent mechanism for influence on the gene expression process [88]. Currently, epigenetics is defined as a reversible and heritable alteration in gene expression that occurs without changes in the primary DNA sequence. Epigenetics can change chromatin and DNA structure via methylation of cytosine bases in DNA, posttranslational modifications of histone proteins, nucleosome remodeling, and non-coding RNAs (including microRNAs) [87] (Figure 1). Epigenetic modifications play a pivotal part in important cellular processes including differentiation, embryonic developmental programming, and the development of cancer. A common feature of tumours is aberrant gene regulation, therefore, epigenetic and genetic changes associated with the initiation and progression of cancer contribute to malignant transformation by working together [89]. Consequently, recent advances in the area of epigenetics contribute directly to our knowledge of cancer progression.

Environmental factors such as pollutants, diet, temperature changes, and other external stresses can modulate the establishment of epigenetic modifications, and thereby influence subsequent gene expression, development, metabolism, and phenotype [90]. DNA methylation, for instance, has a significant role in stem cell differentiation and cellular programming. CpG methylation analysis from pluripotent cell samples (*n* = 269) and somatic cells demonstrated that a distinct methylation signature distinguished hiPSCs (human-induced pluripotent stem cells) and hESCs from somatic cells [91]. As highlighted above there is a range of epigenetic mechanisms including DNA methylation and packaging of DNA by histone proteins or non-coding RNAs [92]. DNA methylation describes the addition of (methylation) or oxidation (hydroxymethylation) of methyl groups, the best known epigenetic markers. These methyl groups are added to cytosine residues in DNA to form 5-methylcytosine (5mC) which can potentially block transcription factor binding access or decrease binding of gene regulatory elements resulting in reduced gene expression [93]. DNA methyltransferase (DNMT) enzymes regulate the initial process of methylation. The three major DNA methyltransferases; DNMT1, DNMT3A, and DNMT3B, drive the methylation pattern of genomic DNA [94]. DNMT1 catalyses the transfer of methyl groups to cytosine nucleotides in CpG islands of genomic DNA. The main function of DNMT1 is to maintain methylation patterns during DNA replication and it has a distinguishable preference for CpGs on hemimethylated DNA [95]. DNMT3A and DNMT3B catalyse de novo methylation of DNA sequences during gametogenesis, embryogenesis and somatic tissue development [96]. The expression level of DNMT3B is low in somatic adult cells while aberrant, elevated, DNMT3B expression is observed in several cancer cells including colorectal carcinoma, hematopoietic cell lines, bladder and breast cancer with the suggestion that DNMT3B expression is required for tumour cell survival [97,98]. DNMT3A and DNMT3B, are expressed at high levels in undifferentiated human embryonic stem cells, but subsequently down-regulated during differentiation [99,100].

Ten-eleven translocation family of dioxygenases (TETs) catalyse the successive oxidation of 5mC to 5-hydroxymethylcytosine (5hmC) [101]. TET family contains three proteins TET1, TET2, and TET3 which can be detected in almost all tissues but are differentially expressed [102]. TET1 and TET2 are the main regulators of 5hmC levels in mouse embryonic stem cells (mESC) [103,104], while elsewhere TET2 and TET3 are expressed more ubiquitously [105]. Loss of TET activity and reduction of 5hmC is associated with a cancer phenotype [106]. Hydroxymethylation levels in mESC are high but decline after differentiation [107].

Histone modification is described by covalent post-translational modification of histones including methylation, phosphorylation, acetylation, ubiquitylation, and sumoylation. They are also responsible for gene expression changes via modification of histone structure or disruption of transcription factor access to promoter sequences. Acetylation of histones is regulated by two main enzyme families; histone acetyltransferases (HATs) and histone deacetylases (HDAC) [108]. HDACs are essential for epigenetic regulation of gene expression, chromosome structure, and control of cellular stability. Their dysregulation is associated with loss of genomic integrity in cancer cells. HDACs remove the acetyl groups on histones which are themselves added by the histone acetyltransferases (HATs) [109]. Specific lysine residues on histones H2B, H3, and H4 are acetylated by HATs increasing DNA accessibility [110]. Histone methylation, a transfer of methyl groups onto constituent amino acids, occurs on the arginine, lysine and histidine residues on the histone proteins [111,112]. Three methylation forms are determined on histone lysine residues: mono-, di- and trimethylation and can be detected using selective antibodies that distinguish methylated histone residues. H3 Lys9 mono- and dimethylation are suggested to be relative to inactive genes in silent euchromatin domains with trimethylation at pericentric heterochromatin [113].

A further contributor to epigenetic modification is non-coding RNAs. These are a cluster of RNAs that regulate gene expression at the post-transcriptional level such as miRNAs (microRNAs), piRNAs (piwi-interacting RNAs), siRNAs (Small interfering RNAs), and lncRNAs (long non-coding RNAs). A number of studies have shown that non-coding RNAs also play an important role in the post-transcriptional regulation of epigenetic-modifying enzymes and genes involved in carcinogenesis [114]. lncRNAs play a role in transcriptional regulation via modification of chromatin structure and DNA methylation levels [88]. lncRNA localization to target loci and subsequent recruitment of chromatin-modifying protein factors can direct chromatin modification [115]. lncRNAs silence gene expression in processes such as X-inactivation or imprinting and also display an enhancer-like role in the activation of gene expression such as KLHL12 [116].

## 4. TERT Expression Regulation by Epigenetic Mechanisms

*TERT* is significant in carcinogenesis and, because of that, is a potential target for cancer treatment. However, the underlying epigenetic regulation of this gene remains ambiguous. DNA methylation, histone methylation–acetylation, and non-coding RNAs all play roles in the regulation of TERT expression in various biological processes including ageing and cancer [18]. The *TERT* promoter harbours a large CpG island (485 CpGs, −1800 to +2300 relative to ATG) within the promoter region, exons 1 and 2 (UCSC Genome Browser). Further, *TERT* promoter and exon 1 were found to be highly methylated in primary acute myeloid leukaemia cell lines [117,118]. DNA methylation is a genome-wide occurrence across CpG islands and non-coding regions, including promoters and enhancers. The promoter regions of genes considered to be highly expressed in cancer are predominantly hypomethylated [119]. The *TERT* promoter is an exception to this model due to its hypermethylated promoter pattern found in most tumour cells and accompanying robust expression [17].

Epigenetic and genetic factors both have roles in determining *TERT* expression levels in tumour cells. For instance, screening of 31 cancer types, including tumours and non-neoplastic tissue samples, revealed that 95% of them displayed telomerase activity and of that 53% displayed altered *TERT* promoter methylation, 31% *TERT* promoter mutations, 3% *TERT* amplification, 3% *TERT* structural variants, and 5% *TERT* promoter structural variants. In addition, *TERT* promoter methylation and mutations were associated with relative telomere shortening when compared to other alterations [15]. Structural 5p15.33 rearrangements, which occur in high-risk neuroblastomas, position the TERT coding sequence adjacent to strong enhancer elements. They are associated with histone modifications, methylation of the rearranged region (including the TERT locus) and high levels of *TERT* expression [120]. *TERT* promoter rearrangements could potentially change the position of enhancer elements which regulate activation of *TERT* expression [15].

Epigenetic mechanisms may be responsible for reversible silencing of the *TERT* during differentiation and stem cell research has the potential to contribute to our knowledge of down-regulation of TERT and other related gene regulations [69,121] Retinoic acid treatment of human embryonal carcinoma and human promyelocytic leukemia cell lines resulted in a decline in telomerase activity during differentiation [122]. hESC differentiation is accompanied by decreased expression of DNMT3A, DNMT3B, and TERT [123]. These indicate that epigenetic regulators, DNMTs and histone methyltransferases could play an important role in the regulation of *TERT* during differentiation. hESCs provide a vital model for the assessment of endogenous epigenetic gene regulation during differentiation in a non-transformed background. Embryonic differentiation contributes to telomeric attrition and the initiation of aging, however, mechanisms underlying telomerase down-regulation remain elusive. Epigenetic studies in ESC differentiation could indicate whether DNA methylation or histone modification contribute to this process and might also help our understanding of TERT regulation in telomerase-positive cancer cells.

## 5. *TERT* Promoter Methylation in Cancer Cells

The *TERT* promoter region includes several GC motifs (72 CpG sites, situated across the region spanning 500 bases UTSS and the first exon) where the promoter methylation pattern affects gene expression [124]. The *TERT* core promoter region spans 260 base pairs including several transcription factor binding sites, e.g., MYC, SP1, HIF1, and WT1 (Figure 2) (Kyo et al., 2008) [67]. Hypoxia (less than 0.1%) induces decreased global methylation in tumour and cancer cell lines [125,126].

The *TERT* promoter consists of a largely hypomethylated CpG island in somatic cells but these are hypermethylated or partially methylated in association with expression and telomerase activation in many cancer cells [127]. The UTSS region of the *TERT* promoter is partially or completely hypermethylated in malignant paediatric brain tumours, which express TERT, while normal brain tissues or low-grade tumours are unmethylated or hypomethylated [128]. Hypermethylation of the *TERT* promoter is associated with tumour progression during the transformation of paediatric brain tumours from low to high grade [128]. The *TERT* hypermethylated oncological region (THOR) located upstream of the *TERT* promoter contains 52 CpG sites across its 433-bp genomic region [129] (Figure 2). Unmethylated THOR suppressed *TERT* promoter activity regardless of *TERT* promoter mutations while hypermethylation of this region increased promoter activity in human tumours [129]. Mechanistically, hypermethylation of the *TERT* promoter reduced the ability of CTCF (CCCTC-binding factor) transcriptional repressor from binding the CCCTC binding region in the first exon of *TERT* (Figure 2) thereby preventing CTCF driven inhibition of *TERT* expression [130]. Treatment of telomerase-positive cells with 5-azacytidine enabled CTCF binding to *TERT* promoter and resultant down-regulation of expression [131]. Consistent with above, sulforaphane, a potent histone deacetylase inhibitor, that also down-regulated DNMTs, drove CpG demethylation and hyperacetylation of the regulatory region (from UTSS, −202 to +106) of *TERT* and promoted binding of MAD1 and CTCF to the *TERT* promoter in MCF-7 and MDA-MB-231 human breast cancer cells [132]. Similarly, the WT1 repressor downregulated *TERT* transcription in a renal carcinoma cell line where overexpression of WT1 suppressed both *TERT* and *MYC* mRNA levels via binding to their promoters [133]. In summary, hypermethylation of specific GpG regions of the *TERT* promoter prevents the binding of transcriptional repressors WT1 and CTCF.

The *TERT* promoter is hypermethylated in a number of telomerase-positive tumour samples while telomerase-negative normal tissues exhibited hypomethylation [17,127]. UTSS hypermethylation is related to lymph node metastasis, aggressive histological features, and poor prognosis in gastric cancer patients [134]. Telomerase-positive human tumour cell lines, as well as tumour tissues from a range of organs, demonstrated the same hypermethylation pattern in the region −441 to −218 from the UTSS containing 27 CpG motifs. This indicated that telomerase activity and *TERT* mRNA levels were strongly associated with the methylation pattern of the promoter region [17]. Renaud et al. demonstrated the partial demethylation of the promoter, between −160 to −80 bp in telomerase-positive tumour cell lines and tumour tissues such as; breast, bladder and cervical cancer [131]. The methylation status of up to 72 CpG sites extending from −500 bases UTSS of the *TERT* promoter was also evaluated across a range of alternate cancer cell lines, with no correlation between specific CpG sites or groups of CpG sites, and *TERT* expression [124]. Further, the distal promoter regions of *TERT*, located between −2056 and −1566 nt, were more methylated in iPSCs than in somatic cells, with a correlation between DNA methylation at the distal region and *TERT* expression levels [135]. Taken together, the promoter region −650 to −200 is generally hypermethylated while −200 to +100 is hypomethylated, partially or hypermethylated (Table 1). These studies also imply that the location of DNA methylation relative to the *TERT* promoter region is important in modifying *TERT* transcription and telomerase activity.

TERT cooperates with the transcription factor SP1 to promote *DNMT3* transcription. Following *TERT* inhibition with siRNA, a substantial decrease in both DNMT3B gene and protein levels indicated that *TERT* promoted DNMT3B expression in HCC, moreover, increased TERT and DNMT3B expression was associated with shorter survival times of HCC patients [54]. All-trans retinoic acid-induced differentiation of HL-60 human leukaemia cells resulted in an inhibition of telomerase activity and changed *DNMTs* expression. Specifically, *DNMT1* expression decreased after 12 days of treatment, while *DNMT3B* expression declined after 6 days of treatment and DNMT3A increased after day 6 to day 12 of all-trans retinoic acid exposure [147]. Further, a sequential reduction in overall DNMT1 expression was noted, and an increase in DNMT1 binding was observed at the *TERT* promoter, indicating maintenance of its hypermethylation status. Genistein, a natural isoflavone, is described as having a capacity for signaling upregulation of tumour suppressor gene expression including P53, P21 and P16 via DNA demethylation and histone modification [148,149]. It also down-regulated telomerase activity and expression of DNMT1, DNMT3A and DNMT3B in breast cancer cells [150]. The inhibition of DNMTs results in hypomethylation of the E2F-1 recognition site at the *TERT* promoter, allowing increased binding (Berletch et al., 2008; Li et al., 2009) [150,151]. Further to above, genistein induced remodeling of the chromatin structures of the promoter by increased H3K9 trimethylation and decreased H3K4 dimethylation [150]. In summary, TERT inhibition is connected to epigenetic modulation of DNMTs and chromatin modifications, in addition, DNMTs are found to be responsible for DNA methylation of the promoter region of the *TERT* [150,151]. A positive feedback loop between DNMT3B and TERT has been suggested through a positive correlation in their expression in a wide range of cancer types within the Cancer Genome Atlas (TCGA) dataset (Yuan and Xu, 2019)[152]. According to their hypothesis, DNMT3B over-expression induces *TERT* promoter hypermethylation, which in turn activates *TERT* transcription, which further up-regulates DNMT3B expression with a positive feedback loop. However, there is not enough data to explain how THOR hypermethylation occurs in cancer. De novo DNA methyltransferase DNMT3B aberrant up-regulation during early cancer progression may induce THOR hypermethylation.

## 6. Transcriptional Regulation of Telomerase by Histone Modifications

DNA is packaged around histone proteins and modifications of DNA and histone tails regulate chromatin structure and gene accessibility, therefore, modifications within the histone core affect the interactions between the nucleosome and DNA [153]. Epigenetic alteration of the *TERT* promoter is strongly linked to its regulation in stem and cancer cells [154]. DNA methylation, histone acetylation, and histone methylation are all vital to the regulation of *TERT* transcription [127,155]. Additional methylation-linked transcriptional control comes via SET and MYND domain-containing protein 3 (SMYD3), a histone methyltransferase, which can bind to the *TERT* promoter and activate *TERT* transcription through histone H3K4 dimethylation or trimethylation in human fibroblasts and cancer cells [156]. Increased *TERT* mRNA levels were observed after overexpression of SMYD3, while its suppression led to reduced H3K4 trimethylation within the *TERT* core promoter and also decreased the ability of MYC and SP1 to bind the promoter in cancer cell lines [157]. Therefore, H3K4me3 on the *TERT* promoter through SMYD3 provides an optimal binding condition for MYC and SP1. H3K4me3 is considered a positive marker for gene activation.

Methylation of histone tails, occurring on lysine and arginine residues, may induce repression or expression of genes related to methylation level and methylated residue [158]. Mono and dimethylation of H3K9 within promoter regions are associated with inactivated gene expression whereas acetylation of H3K9 is related to open chromatin and gene transcription [18].

Common histone modifications in the *TERT* promoter are H3K27ac and H3K4me3 active histone marks and epigenetic silencing H3K9me3 and H3K27me3 marks (Table 2) [149,159]. H3K4me3 associated with active transcription is significantly enriched in cancer cells and iPSC [160] whereas the H3K27me3 and HP-1α forming heterochromatin are significantly enriched in somatic cells [135]. As previously mentioned, *TERT* promoter mutations exhibit H3K4me2/3, an active chromatin mark in multiple cancer cells [76]. Akincilar et al. showed that *TERT* promoter region mutations (−146C>T and −124C>T) exhibit increased H3K4me3 and H3K9Ac marks, where CRISPR mediated reversal of mutant to wild-type *TERT* promoter leads to a reduction of telomerase activity and active histone marks [161].

Zinn et al. reported that the *TERT* promoter region spanning −150 to +150 from UTSS displayed active chromatin remodelling and an overall lack of methylation, enabling transcription of the *TERT* gene. Nevertheless, the −600 bp UTSS region is intensely methylated in breast, lung, colon cancer cells and immortalized cell lines [136]. Additionally, they observed both H3K9ac and H3K4me2, active chromatin signs, and H3K9me3 and H3K27me3, methylated inactive DNA, in the *TERT* promoter [136]. H3K4 hypoacetylation and H3K9 methylation diminish *TERT* expression in telomerase-negative cell lines, while on the other hand, H3 and H4 hyperacetylation and H3K4 methylation are associated with *TERT* transcription in telomerase positive cells [162,163].

*TERT* promoter activity is decreased 5–10 fold after 14 days differentiation of hESCs [159]. Additionally, histone markers H3K4me3 and H4Ac were observed in hESC-(wt) containing single-copy bacterial artificial chromosome reporters, which included a 160-kb human genomic sequence comprising the whole *TERT* locus. However, during differentiation, these active histone marker levels were considerably down-regulated [159]. It was concluded that H3K4me3 and H4Ac have significant roles in epigenetic regulation, and are associated with *TERT* promoter activities in hESCs and differentiated cells. Moreover, a considerable increase in H3K9me3 and H3K27me3 levels on *TERT* promoter are observed and these two marks are related to gene silencing in differentiated hESCs [159].

Histone deacetylation is a repressive feature for *TERT* expression in normal cells [159,164]. Inhibition of HDAC accelerated TERT protein degradation, decreased telomerase activity and induced the senescence process in rat vascular smooth muscle cells [165]. Consistent with above, telomerase activity and *TERT* expression were downregulated in brain cancer cells A-172 (glioblastoma) and ONS-76 (medulloblastoma) after exposure to HDAC inhibitors [166]. Alternatively, Zhu et al. demonstrated cell proliferation and apoptosis induction after histone deacetylase inhibitor exposure in ovarian cancer cells, but telomerase activity stayed at the same level [167]. As a result, the effect of HDAC inhibitors on telomerase activity and TERT expression may be best understood as being cell-type or transformation-level-associated.

**Table 2 cancers-13-01213-t002:** Histone modifications of the *TERT* promoter.

Histone Modification	Associated Mechanism	Cell Type	Reference
H3K4Me3, H3K9Ac, H3Ac H4Ac (Active marks) H3K9Me3, H4K20Me (Repressive)	The lowest expression of repressive marks but the highest levels of active marks in the cell lines expressing high levels of TERT Higher levels of repressive marks at the promoter in ALT cell lines	WI38, SUSM-1, KMST-6, and WI38-SV40 (ALT cells lines) C33a, A2780, and 5637 (high TERT expression)	[163]
H3K4Me2/3 (Active marks)	Increase occupancy of MYC and SP1 on the *TERT* promoter	HCT116 and L1236 cells	[156]
H3K9Ac, H3K4Me2 (Active marks) H3K9me3, H3K27me3 (Repressive)	Both active and inactive chromatin signs present across the *TERT* promoter	RKO, SW480, HCT 116, MCF7, and VA13 cells	[136]
H3K9Ac (Active marks) H3K9Me3 (Repressive)	Depletion of SIRT1 induces H3K9Ac and reduce H3K9Me3 at *TERT*	Hepatocellular carcinoma	[168]
H3K4Me2/3 (Active marks) H3K27me3 (Repressive)	Mutant *TERT* promoters exhibit the H3K4me2/3 and recruit the GABPA/B1 transcription factor in multiple cancer cell lines, while the wild-type allele exhibits the H3K27me3	HepG2, SNU-475, SNU-423, UMUC3 and T24	[76]
H3K4Me3, H3K9Ac (Active marks)	Enrichment of both H3K4Me3 and H3K9Ac in the proximal *TERT* promoter region in mutant cell lines	BLM, A375, T98G (bearing -146 C>T mutation), and U251 (bearing −124 C>T mutation)	[161]
H3K4Me3 (Active mark)	Increased H3K4Me3 allows MYC/MAX binding to E-boxes in the *TERT* promoter	E6E7 IDH1mut pre- and postcrisis cells	[160]
H3K27Me1, H3K27ac (Active marks) H3K27me3 (Repressive)	Enrichment of active marks in the juxtaposed to the *TERT* promoter	NIH Roadmap Epigenomics Consortium (111 epigenomes)	[15]
H3K4me3, H4Ac (Active marks) H3K9Me3 H3K27me3 (Repressive)	*TERT* silencing during differentiation accompanied by increases of repressive marks	Human embryonic stem cell	[159]
H3K4me3 (Active mark) H3K27me3, HP-1α (Repressive)	H3K4me3 is enriched in iPSCs, whereas H3K27me3 and HP-1α levels are higher in somatic cells	Fetal lung fibroblast (MRC) and MRC-iPSCs	[135]
H3K4Me3, H3Ac (Active marks) H3K27Me3 (Repressive)	Active marks are enriched at *TERT* core promoter	Acute promyelocytic leukemia cells NB4-LR1 and NB4-LR1	[169]

## 7. Regulation of TERT via Non-Coding RNAs

Non-coding RNAs are functional RNA molecules transcribed from DNA that modulate gene expression at transcriptional and post-transcriptional levels. miRNAs are 18–25 bp non-coding RNAs. They contribute to numerous biological processes, including carcinogenesis, by acting as tumour suppressors or oncogenes [170]. A large number of studies have identified miRNAs as potential biomarkers for human cancer diagnosis, prognosis and as therapeutic targets [171,172]. Usually, miRNAs bind to recognition sites in the 3′ untranslated regions (3′UTRs) of mRNA transcripts, specific base-pair sequences within the 5′UTRs and open reading frames (ORFs) [173,174,175]. Several non-coding RNAs interact with TERT. Multiple miRNAs target the 3′-UTRs and ORFs of TERT, regulating its activity in several cancer cells (Table 3). For instance, miR-491-5p functions as a tumour suppressor and suppresses cell growth by directly targeting 3’ UTR of TERT mRNA in cervical cancer [176]. TERT is directly controlled by miR-128 and miR-138 in different cancers [177,178]. miRNAs can modulate transcription through inhibition of transcription factors of TERT. For example, miR-21 represses TERT through a STAT3 transcription in glioblastoma cells and regulates TERT via the PTEN/ERK1/2 signaling pathway in colorectal cancer [59].

LncRNAs modulate transcriptional regulation by acting as scaffolds to facilitate the assembly of specific transcriptional complexes or post-transcriptional regulation via serving as sponges to reduce the functional availability of specific miRNAs [179,180]. lncRNAs can also provide competitive target binding to miRNAs and prevent their binding to target RNAs [181]. For example, BC032469, a highly expressed lncRNA in gastric cancer, binds to miR-1207-5p (Table 3.) and upregulates TERT expression and proliferation [182]. Further circWHSC1 (circular RNA) sponges miR-1182 (Table 3.) and miR-145, resulting in increased expression of TERT and progression of ovarian cancer [183]. Taken together, these indicate that there are alternate epigenetic modifiers that directly or indirectly regulate TERT. In addition, lncRNAs can change epigenetic modifications or chromatin modelling to influence alternative splicing and lncRNAs generation [184].

**Table 3 cancers-13-01213-t003:** Regulation of TERT through miRNAs.

miRNAs	Tissue Type	Mode of Action	Assay	References
let-7g *	Idiopathic pulmonary fibrosis	Interaction with 3′UTR of TERT to reduce telomerase expression	Reporter Assays	[185]
miR-21	Glioblastoma	Decreases TERT through a STAT3-dependent manner	Reporter Assays and CHIP	[59]
miR-128	HeLa, lung cancer, colon cancer and pancreatic cancer cells	Interacts directly with the coding sequence of TERT and reduces the mRNA and protein levels	Reporter Assays, Immunopurification strategy	[178]
miR-133a, miR-342, miR-541	Jurkat cells	Interaction with the 3′UTR of TERT to reduce expression	Reporter Assays	[186]
miR-138	Thyroid carcinoma cell lines	Interacts with 3′UTR of TERT and reduces protein expression	Reporter Assays	[187]
miR-498	Ovarian and breast cancers	Targets the 3′UTR of TERT mRNA and decreased its expression	Ribonuclease Protection Assays, Reporter Assay	[188,189]
miR-1182	Bladder, ovarian and gastric cancers	Targets 3’UTR and ORF1 of TERT and reduces protein expression	Reporter Assay Western blotting	[190,191,192]
miR-1207-5p, miR-1266	Gastric cancer	Interacts with the 3’ UTR of TERT and suppresses TERT	Reporter Assay	[193]
miR-497-5p, miR-195-5p, miR-455-3p	Melanoma cells	Targets TERT and decrease expression at both mRNA and protein levels	Dual-luciferase reporter assay	[194]
miR-155	Bladder cancer	Association of lower expression of miR-155 and higher expression of TERT	RT-PCR	[195]
miR-512-5p	Head and neck squamous cell carcinoma	Targets 3’ UTR and downregulates *TERT* expression and telomerase activity	Dual-luciferase reporter assay	[196]
miR-661	Glioma cells	Targets 3’ UTR and silence *TERT*	Reporter Assay	[197]
miR-491-5p	Cervical cancer	Targets 3’ UTR of TERT and decreases protein expression	Dual-luciferase reporter assay	[176]

let-7g * (let-7g-3p).

## 8. Conclusions

The potential clinical application of TERT expression and telomerase activation or telomerase activity inhibition remains at the forefront of a number of cancer therapeutic strategies. Regulation of telomerase via epigenetic modification aids further understanding of the onset of cancer and holds significant potential for therapy. Epigenetics and genetics in concert may provide the necessary framework to resolve the remaining challenges. The continued development of an understanding of TERT regulatory mechanisms will help clarify this complex, multifactorial, process. It is clear that alternate control mechanisms work in a synchronized and orchestrated manner to ensure proper regulation of telomerase activity and maintenance of telomere lengths. Epigenetic manipulation of TERT could be a promising approach for the treatment of cancer.

The hypermethylation pattern of *TERT* promoter is unique to human cancer and might be useful as a cancer diagnosis [129,198] and also as a prognostic biomarker [128] but further research, including large cohorts of patients, is required for future clinical application. The *TERT* promoter is an attractive target for tumor-specific therapy to selectively kill cancer cells without impacting somatic cell populations due to high levels of telomerase expression in cancer cells [199]. The *TERT* promoter is a potential target for epigenetic-driven telomerase therapy. By combing dCas9 (catalytically dead Cas9) with the Tet1 (Ten-eleven translocation methylcytosine dioxygenase 1) catalytic domain, an enzyme involved in regulating the demethylation of DNA, researchers were able to successfully demethylate a region of DNA and subsequently upregulate the transcription of the target genes [200]. The application of dCas9 could represent an efficient mechanism for the modification of target DNA methylation. As yet, no studies describe the application of CRISPR dCas9 in the demethylation of the THOR to reduce telomerase activity in cancer. A possible application in cancer therapy is using CRISPR dCas9 to demethylate the *TERT* promoter region to reduce telomerase activity. G-quadruplexes (G4s) are important for transcription and replication processes due to their role in both DNA and histone methylation. G4 structures can act as an obstacle to replication fork progression and disrupt histone recycling, which means that original histone modifications can be replaced with new histones resulting in epigenetic reprogramming [201]. The *TERT* promoter has a high propensity of G4-formation [202]. The linkage of G4s to the epigenetic state of the *TERT* promoter is an emergent concept and the clinical utility remains to be determined. It is suggested that CpG methylation at the *TERT* promoter induces G4 formation, enriched near *TERT* TSS, that results in disrupted CTCF protein binding and an increase in transcription [203].

The current diversity of epigenetic tools available creates an opportunity for novel strategies for reversible telomerase regulation and targeted telomere maintenance pathways. For example, preclinical and clinical information has suggested that histone deacetylase inhibitor (HDACi) can create a new therapeutic concept for biomedical applications [204]. HDACi mediated telomerase intervention is an example of an epigenetic focused therapy. HDACi reactivated transcriptionally silenced apoptosis via both the extrinsic and intrinsic pathways, and induced tumour suppressor gene expression including P21 and P53 [205]. Increased expression of HDAC1, HDAC2, HDAC3 and HDAC6 are reported in several cancers including colon, prostate and breast cancer and inhibition of HDACs potentially could target proliferation, differentiation, angiogenesis, and migration [206]. Mechanistically, the inhibition of HDACs increased acetylation of histone and non-histone proteins, leading to an increase in transcriptionally active, open, chromatin. The Food and Drug Administration (FDA) has approved several HDAC inhibitors as anti-cancer drugs including Vorinostat [207], Romidepsin [208,209], Belinostat [210], and Panobinostat [211]. One attractive candidate for HDACi-mediated inhibition is telomerase. Vorinostat downregulated *TERT* and repressed telomerase activity in non-small cell lung cancer and lymphoma cells [212,213]. Vorinostat downregulated DNMTs (DNMT1 and DNMT3B) and induced demethylation on the *TERT* promoter [212]. HDACi inhibited telomerase activity and downregulated TERT in childhood brain tumour cells [214], human normal TERT-immortalised fibroblasts, brain cancer cell lines [166], small-cell lung cancer lines [215] and prostate cancer cells [216]. In contrast, HDAC inhibitor trichostatin A treatment induced significant expression of *TERT* mRNA and telomerase activity in somatic cells, while, under the same conditions no altered response was noted with cervical cancer cell lines [164]. Further, Choi et al. reported that trichostatin A repressed *TERT* expression via demethylation and recruitment of CTCF into the demethylated *TERT* promoter region in colon cancer cells [217]. It remains to be determined why the colon cancer cell line response was distinct from other studies but highlights that a single model of activity for trichostatin A activity may be premature. Moreover, HDAC9 promoted ALT pathway by increased formation of ALT-associated promyelocytic leukemia nuclear bodies in ALT positive cells [218]. Simultaneous targeting of ALT and telomerase positive cells with HDACi may present a viable, dual mechanism for cancer therapy.

Although telomerase possesses desirable advantages as a target, there are challenges and limitations for the development of successful clinical therapies. For example, targeting telomerase alone requires an extended treatment time to see gradual attrition of telomeres. However, a combination with other targeted therapies might be a solution to overcome challenges and achieve optimal anticancer effects. Moreover, the outcomes of targeting TERT expression are expected to be different from those arising upon the inhibition of the telomerase catalytic activity alone.

## Figures and Tables

**Figure 1 cancers-13-01213-f001:**
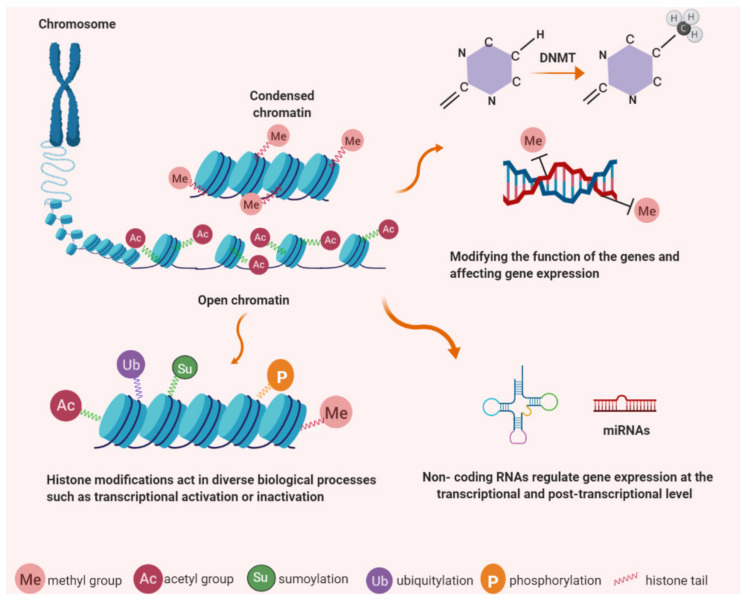
Epigenetic modifications have important roles in chromatin structure and gene expression. DNA is wrapped around histone proteins to form chromatin; less condensation associates with transcriptional activity while more condensed chromatin is found in a state of transcriptional silencing. Epigenetic mechanisms occur as covalent modifications of either DNA or histone proteins. DNA methylation (e.g. cytosine methylation and hydroxymethylation) and histone modification, (e.g. acetylation, methylation, phosphorylation, ubiquitination and sumoylation). Non-coding RNAs regulate protein-coding genes and associate with DNA methylation and histone modification.

**Figure 2 cancers-13-01213-f002:**
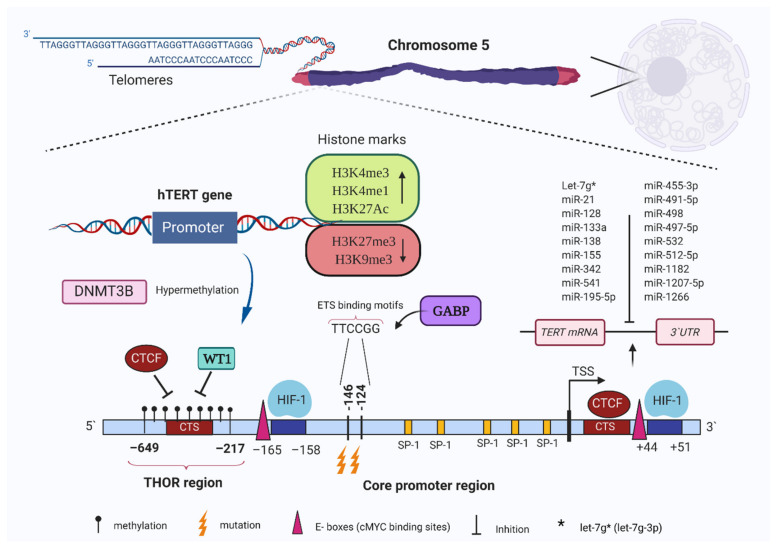
Human telomerase reverse transcriptase (*TERT*) promoter region and binding sites for various transcription factors, repressors, and repressive chromatin markers. The specific mutations relative to ATG and hypermethylated regions relative to upstream of the transcription start site (UTSS) (*TERT* hypermethylated oncological region—THOR) are represented. Cancer-specific mutations control the recruitment of GABP transcription factor to the promoter and increase gene expression. Methylation of promoter potentially controls transcriptional activator (MYC) and repressor (WT1 and CTCF) recruitment to target sites, leading to high expression. Green box active histone marks, red box repressive marks. Black dots show methylated cytosines followed by guanine residue (CpG) sites. ETS: E-twenty-six; TSS: transcription start site.

**Table 1 cancers-13-01213-t001:** Reported methylation status of the *TERT* promoter region.

Cell Types	Region Tested	Methylation Status	Reference
SUSM-1, CMV, SiHA, MDA-231/435, Calu 1/3/6, HTB 57/178/182, SW480, HTC 116 (Telomerase +)	From −500 UTSS to +50 first exon (72 CpG sites)	Partial or total methylation	[124]
A549, HTB183 (Telomerase +) NHF, MRC-5/p27 (Telomerase −)	From −500 UTSS to +50 first exon (72 CpG sites)	Unmethylated	[124]
U2OS, telomerase-negative breast carcinoma, VA13 GM847	Promoter region	Partial or total methylation	[127]
CT1485 (Telomerase +) WI38, HA-1 pre-crisis cell strain, JFCF-6T/5K pre-crisis cell strain, IMR90, BJ fibroblast, adrenal carcinoma (Telomerase −)	Promoter region	Unmethylated	[127]
J82, T24, MCF-7, A431, HeLa, Co115, HT29, SW480, H520, SW2, PC-3,	27 CpG sites −441 to −218 relative to UTSS	Partial or total hypermethylation and correlation with *TERT* mRNA expression and telomerase activity	[17]
Tumour tissues from brain, breast, bladder, colon, kidney, lung, soft tissue			
Saos-2, U2-OS (Telomerase −)	27 CpG sites −441 to −218 relative to UTSS	Partial or total hypermethylation	[17]
Normal tissues; skin, brain, bladder, muscle, kidney, heart, placenta, testis, colon		Unmethylation and telomerase negative	
HeLa, SW480, Tumour tissues (breast, bladder and cervix) (Telomerase +)	−100 to +100	75 to 100% methylation	[131]
	−165 to −100	0 to 55% methylation	
BJ fibroblasts (Telomerase −)	−165 to +100 relative to UTSS	3 to 23% hypomethylation	[131]
Caco-2, HCT116, RKO, SW480, MCF7, MDA-MB-231, MDA-MB-435S, MDA-MB-453, H82, H157, H209, H146, H358, H417, H549, H747, H1299, U1752, DMS53, HL-60, KG-1a, Jurkat, Raji, LCL	−600 bp relative to UTSS	Partial or total methylation and TERT expression	[136]
	−150 to +150 relative to UTSS	Partial or unmethylation	
Hepatocarcinoma cell lines HepG2, SNU-182, SNU-398, HCC Tissue (Telomerase +)	−165 to +49 relative to UTSS	10 HCC clones and HepG2 Hypo/unmethylated 12 HCC clones and SNU-182, SNU-398 hypermethylated and reduced TERT	[137]
Normal liver tissue (Telomerase −)	−165 to +49 relative to UTSS	Hypermethylated and reduced TERT	[137]
Malignant pediatric brain tumours	5 CpG sites located UTSS	Hypermethylation TERT expression	[128]
Neuroblastoma	UTSS	Highly methylated	[120]
Hepatocellular carcinoma	−452 to −667 and −974 to −1419 relative to UTSS	Hypermethylation and high TERT expression	[138]
Gastric cancer	25 CpG sites located −555 from UTSS	Hypermethylation and high TERT expression	[134]
Haematopoietic cell lines including Jurkat, THP1, K562, cervical cancer and embryonic kidney cells	−60 to +20 relative to ATG	Hypermethylation and TERT expression	[117]
Normal peripheral blood cell populations (granulocytes, T cells, B cells and monocytes)	−60 to +20 relative to ATG	Unmethylated	[117]
Acute myeloid leukaemia, myelodysplastic syndrome	−520 to −400 relative to ATG	Hypermethylation	[117]
Thyroid cancer	−541 to −578 relative to ATG	Hypermethylation	[139]
Prostate cancer	52 CpG –140 to –572 relative to the UTSS	Hypermethylation	[140]
Melanoma	26 CpG –482 to –667, relative to the ATG	Hypermethylation and positively correlated with the TERT expression	[141]
Melanoma	−945 to −669 bp relative to UTSS	Hypermethylation and positively correlated with the TERT expression	[142]
Pancreatic cancer	Position –575 relative to the UTSS	Hypermethylation	[143]
Colon, blood, breast, prostate, brain, lungs, bladder, ovaries, bone, skin cancers	52 CpG sites −100 to −600 located UTSS	Hypermethylation and upregulated TERT expression	[129]
Bladder cancers	5 CpG sites in –140 to –572 region relative to UTSS	Hypermethylation and higher TERT expression	[132,144]
Primary metastatic medulloblastoma	5 CpG sites localized −600 bp UTSS region	Hypermethylation	[145]
Thyroid Cancer Cell Lines	−662 to +174 relative to UTSS	Hypermethylation of upstream promoter and correlated with TERT expression	[146]

## Data Availability

Data available in a publicly accessible repository.

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
