# Peer review of "Telomerase Regulation: A Role for Epigenetics"

_cancers, 2021, doi:10.3390/cancers13061213_

Round 1

Reviewer 1 Report

The manuscript contributed good to the related fields.

Author Response

We thank the panel of reviewers for their constructive feedback.  Please see the revised version of the manuscript.

Reviewer 2 Report

The manuscript focuses on the role that epigenetics can play in regulating the expression of telomerase. The authors have improved their manuscript, but there are several points that need to be revised.

COMMENTS

-Title: The question mark in the title is not appropriate: it is well known that epigenetic is involved in TERT expression.

- Overall, the manuscript still contains several repetitions and inaccuracies.

-Abstract, lines 26-27: …“a further role in therapeutic resistance in human stem cells, breast and cervical cancer..”. The therapeutic role has not been studied only in these tumors, the sentence needs to be reformulated.

-Page 4, lines 179-181. The sentence “….telomerase activation in tumor cells, but ….histone modifications”. Histone modifications concur to the gene transcription.

-Page 5, lines 219-225.  The role of TERT promoter mutations needs to be more clearly defined.

-Page 12, lines 434-436: The sentence is unclear.

-Page 12, line 438:    “while” does not seem appropriate.

-Page 13:  The two sentences (lines 515-517 and lines 517-518) are consistent with each other. Why “On the other hand”?

-Page 17, lines 619-621. There is a mistake:  HDAC inhibitors instead of HDACs may be useful to simultaneously target ALT and telomerase positive cancer cells.

-All genes need to be indicated following the HUGO gene nomenclature guidelines (for example, MYC and not c-Myc, see page 5 etc.).

Author Response

We thank the panel of reviewers for their constructive feedback. We have detailed in the attachment the manner, point by point, in which these have been addressed. Please see the attachment and also the revised version of the manuscript.

Reviewer 3 Report

Though the topic is pretty sounds and scientifically important, the way it has been faced is rather superficial and inaccurate. The manuscript results pretty unfocused and redundant, whereas take home messages remain unclear. Conclusions are pretty scanty and the paper lacks of authors’ own criticisms.

Major concerns:

-The simple summary is uninformative. The notions reported within each sentence must be logically connected to each others. In my opinion, the main concept to be reported at lines 7-8 should be that telomere maintenance, which may be accomplished by the reactivation of telomerase activity, is a fundamental step in carcinogenesis. Moreover, what kind of information is delivered in the sentence at lines 11-13? Did the authors mean that DNMT and HDAC inhibitors used in the clinics have a proven effect in terms of inhibition of telomerase activity?

-Lines 31-34: On the basis of which evidence the authors may state that epigenetic alterations play a key role in the deregulation of TERT expression in cancer cell lines. Saying that cancer cell lines expressing TERT show 53% of methylation and 31% of mutations does not appear to be a solid argument to sustain that TERT expression in cancer is prominently driven by epigenetic rather than genetic alterations (compare lines 346-357).

-Lines 44-138: Basic concepts on telomere biology, telomere structures and telomerase have been described in a very rudimentary manner. The entire section is characterized by several redundancies and inaccurate information has been mainly delivered. It needs to be entirely re-written: please be focused; only useful information must be delivered and the different concepts must be logically connected to each others.

-Lines 55-56: Are Ku proteins the only ones to be associated with telomerase? What kind of information is delivered in this sentence?

-Lines 59-61: which is the logical connection between these two sentences?

-Lines 64-67: The sentence is completely out of context. Please delete it.

-Line 69: The beginning of this sentence (Telomeres themselves are…telomere repeats…) is a circular reasoning.

-Lines 93-96: there is no logic connection between these sentences.

-Lines112-120: this part on ALT mechanism has nothing to do with the epigenetic control of TERT expression. This part must be removed, unless the authors are willing to dig and accurately report on the basics of ALT, its role in cancer and how it is controlled by epigenetics.

-Lines 179-181: This sentence is a circular reasoning (Transcriptional regulation of TERT is significant for telomerase activity…but…DNA and histone modifications…might also be important regulators of TERT transcription).

-Lines 237-239: The sentence is unclear. What does “…interference via dominant negative competitive forms” mean? Could the authors better report on the biological evidence that the expression of the three quoted TERT splicing variants decreases telomerase activity?

-Lines 246-248: Which is the logical connection between this sentence and the previous one?

-Lines 336-337: Please provide details on the genomic region spanned by TERT promoter (see lines 373-377).

-Lines 343-345: These sentences are misleading, as TERT is not an oncogene.

-Lines 550-567: As there is no evidence the TERRA act as regulator of TERT expression via an epigenetic mechanism, this part of the manuscript must be removed. The evidence that TERRA is an lncRNA has nothing to do with the epigenetic control of TERT transcription.

-Lines 591-596: The part of the manuscript regarding a possible role of G-quadruplex in the epigenetic control of TERT transcription sounds merely appended. Moreover, the logical connection between the role of G4 in transcription/replication and DNA/histone methylation is pretty elusive. This topic must be better put into the context and properly discussed.

-Line 600: Which is the relationship between HDAC inhibitors and RNA targeting?

-Line 601-610: This information is unnecessary. The evidence that FDA has approved different HDACi for cancer therapy has nothing to do with the epigenetic control of telomerase expression by these inhibitors. Instead, the authors must discuss whether there is clinical evidence that these drugs may impact on TERT expression or whether available evidence, if any, are still at pre-clinical level.

-Line 616: please explain what does “…overexpression of Sp1 enhances responsiveness” refer to.

-Line 614-615: The authors should reconcile the evidence that TSA induced significant expression of TERT mRNA and telomerase activity in normal cells but not in cancer cells with the findings reported by Choi JH et al. (BBRC 391:449, 2010) that the drug repress the expression of TERT in cancer cells.

-lines 619-621: this information is not necessary, unless the authors are willing to dig and accurately report on the basics of ALT, its role in cancer and how it is controlled by epigenetics.

-Lines 633-637: this part of the text needs to be better framed. The notion that telomerase inhibitors would require a lag time to exert a therapeutic effect, consequently they are not suitable as single agents for anticancer therapeutic regimens, is elusive for the broader readership if not put into the context of the fundamentals of telomere/telomerase biology. In addition, while the requirement for a lag time may be necessary for the inhibitors of telomerase catalytic activity, the same might not hold true in the case of agents targeting TERT, which has been clearly documented to exert an extra-telomeric, pro-survival role in cancer. Consequently, the outcomes of interfering with TERT expression are expected to be remarkably different from those arising upon the inhibition of the telomerase catalytic activity.

-Table 3: a column showing the assays used to biologically validate TERT as a direct target of the listed miRNAs must be provided.

-The list of reference must be carefully checked for redundancies (see for instance, ref n. 78 and 79).

-Language needs extensive editing.

Minor concerns:

-Lines 7-9: The sentence has no meaning. It should be deleted.

-Line 45: Please delete (RNA-protein): the word “ribonuleoprotein” is self-explanatory.

-Line 61: it is more correct to write “gene copy number variations”. Please adjust.

-Line 62: histone acetylation and methylation must be included in this short list of epigenetic modifications.

-Lines 140-178: The sequential numbering of the manuscript sections must be adjusted.

Author Response

We thank the panel of reviewers for their constructive feedback. We have detailed corrections in the revised version of the manuscript.

Reviewer 4 Report

The manuscript has been largely improved and reads much better. I noted several issues that remained and should be corrected.

  • Line 220. Contrary to what the Authors state the aa-like mode of mutation designation (C250T) remained. Nucleotide positions should be given as -146 etc and indicated as C>T as in the original papers! Also, please, note these nucleotide positions were given relative to ATG in the cited references.

  • In Table I nucleotide positions are given as relative to TSS, ATG, UTSS or ..nothing. This should be unified into a consistent style e.g. relative to TSS or ATG (and the position of ATG relative to TSS should be given in figure legend). Alternatively a simple scheme showing the promoter region with the positions of TSS and ATG indicated should be added.

  • If TERT is written in italics the word “gene” should be omitted, especially that two different versions: TERT gene promoter and TERT promoter can be found in the text.

  • Line 418; skip “hypermethylated”

  • Line428: UTSS -650 to “-“200

  • Line 28: TERT instead of TERT? Meaning: activation or silencing of TERT gene.

  • It seems as if hTERT was removed from the text but remained in the Abstract and appears again in Conclusions, is this correct?

  • Line: 434 -439. In red -not clear whose impact on TERT promoter remains undetermined? The level of DNMT3B certainly remains the same.

  • Line 451: “genistein induced” rather than “genistein-induced”?

  • Line 484: histone acetylation

  • Line 591: region(s) twice in one sentence.

Author Response

(The authors gave the same response as above.)

Round 2

Reviewer 2 Report

The revision  made  by the authors has improved the manuscript

Only few minor comments:

- lines 173-175: -124 and -146 bp positions for TERT promoter mutations refer to ATG start site and not to the GC boxes.

- lines 177-178: this sentence is a repletion of the previous one.

-lines 135: “also” is not appropriate and should be removed.

- the gene nomenclature need  to be revised. Check all the gene nomenclature throughout the manuscript.

Author Response

We thank the panel of reviewers for their constructive feedback. Please see the revised version of the manuscript and cover letter attached.

Reviewer 3 Report

The authors have  replied to the reviewer's concerns . Editing of English language and style is recommended.

Author Response

We thank the panel of reviewers for their constructive feedback.  English in the manuscript thoroughly checked and edited for language. Please see the revised version of the manuscript.

This manuscript is a resubmission of an earlier submission. The following is a list of the peer review reports and author responses from that submission.

Round 1

Reviewer 1 Report

The manuscript is well written. However, some points are needed to be revised.

1. In Fig. 1 and line 282-287, it is not clear how non-coding RNAs regulate gene expression.

2. In the conclusion, what is the possible application of epigenetic regulation on telomeres regulation for cancer therapy, particular in cancers that are ALT?

Author Response

We thank the panel of reviewers for their constructive feedback. We have detailed in the attachment the manner, point by point, in which these have been addressed. Please see the attachment.

Point 1. In Fig. 1 and line 282-287, it is not clear how non-coding RNAs regulate gene expression.

Response 1: To address the points detailed above we have inserted and modified the following text.

Page 7. Lines 325-330.

`lncRNAs play a role in transcriptional regulation via modification of chromatin structure and DNA methylation levels 88. Long non-coding RNA localization to target loci and subsequent recruitment of chromatin modifying protein factors can direct chromatin modification 115. lncRNAs are evidenced to silence gene expression in processes such as X-inactivation or imprinting and also display an enhancer-like role in the activation of gene expression such as KLHL12 116.’

Page 14. Lines 539-541.

`LncRNAs modulate transcriptional regulation by acting as scaffolds to facilitate the assembly of specific transcriptional complexes or post-transcriptional regulation via serving as sponges to reduce the functional availability of specific miRNAs 181,182.`

Page 15. Lines 548-566.

`Non-coding RNAs are functional RNA molecules that are not translated into protein but instead target epigenetic modifying enzymes, or transcription factors, to regulate gene expression. Telomeric repeat-containing RNA (TERRA) is an example of non-coding RNA that also plays a role in telomere maintenance mechanisms (TMM). TERRA, transcribed by RNA polymerase II from telomeres, localizes at chromosome ends and plays a role in heterochromatin formation and the regulation of telomerase activity at mammalian chromosome ends in both human and mouse models 188 189. A role for TERRA as a general inhibitor of telomerase activity remains controversial. TERRA has the same sequence complementarity to the telomerase template and in vitro studies demonstrated that TERRA acts as a competitive inhibitor of telomerase 188 190. However, TERRA overexpression in mammalian cells is accompanied by unaltered telomerase activity and efficient elongation of telomeres, suggesting a lack of telomerase activity inhibition 191 192. TERRA-telomerase interactions may be more complex than simply regulating telomerase activity. Interestingly, TERRA appears more amplified in ALT cells than in telomerase-positive cells in general 193. It remains to be determined what the precise role of TERRA is in ALT and telomerase positive cells and whether they provide the same functionality.’

Point 2. In the conclusion, what is the possible application of epigenetic regulation on telomeres regulation for cancer therapy, particular in cancers that are ALT?

Response 2: In response to the above point we have added the following text.

Page 16. Lines 597-600.

`The current diversity of epigenetic tools available creates an opportunity for novel strategies for reversible telomerase regulation and targeted telomere maintenance pathways. For example, preclinical and clinical information has suggested that histone deacetylase inhibitor (HDACi) and RNA targeting create a new therapeutic concept for biomedical applications 212`.

Page 17. Lines 618-621.

`Moreover, HDAC9 promoted ALT pathway by increased formation of ALT-associated promyelocytic leukemia nuclear bodies in ALT positive cells 225. Simultaneous targeting of ALT and telomerase positive cells with HDACs may present a viable dual mechanism for cancer therapy.`

Reviewer 2 Report

The authors present an overview of the epigenetic mechanisms involved in the regulation of TERT expression/telomerase activity. The field is of interest, but the manuscript should be revised

MAJOR COMMENTS

-Title: The question mark in the title is not appropriate: it is well known that epigenetic is involved in TERT expression.

- Overall, the manuscript contains many repetitions and several inaccuracies.

  1. i) Sections 1-4: these sections are practically introductory to the central topic of the review. They contain a superficial summary of TERT characteristics. Repetitions should be eliminated, and the text should be more focused on the main topic. There are several inaccuracies, for example:

1) Page 4, lines 191-194: the mutation sites -124 (C228T) and -146 (C250T) cannot be called polymorphisms. Both are mutations and should be discussed appropriately.

2) Page 4, lines 188-190: the complex MYC and MAX activates the TERT promoter (Takakura et al, 1999), not the complex MYC and MAD1.

3) Page 5, lines 199-202: the splice variant which  produces the biologically functional protein  contains both α and β regions.

  1. ii) Sections 5-8: the roles of methylation, acetylation, and non-coding RNAs in the epigenetic control of the TERT transcription are not clearly presented.

4) The authors should detail the role of hypermethylation of TERT promoter resulting in the increase of TERT transcription. This aspect deserves particular attention.

5) Page 11, lines 379-380: the sentence is not in accordance with the paper of Yu et al., 2018: TERT cooperates with the transcription factor SP1 to stimulate DNMT3B transcription, but the TERT promoter was not investigated.

6)The role of histone acetylation on TERT transcription is really confused and contradictory; it needs to be rewritten.

7) The conclusions should be more focused, taking into account the most important and consistent findings about the role of epigenetics in TERT expression

8) Page15 lines 502-519: the role of TERT as a target candidate for HDACi should be significantly revised and better defined

MINOR COMMENTS

All genes need to be indicated following the HUGO gene nomenclature guidelines (for example TERT and not hTERT, MYC and not c-Myc).

The acronym for TRF2 on page 1, line 42 needs to be provided the first time the gene is named.

On page 8, line 341, UTSS stands for upstream transcription start site

The manuscript contains several grammar errors and typos.

Author Response

We thank the panel of reviewers for their constructive feedback. We have detailed in the attachment the manner, point by point, in which these have been addressed. Please see the attachment and also revised version of the manuscript.

Point 1. Title: The question mark in the title is not appropriate: it is well known that epigenetic is involved in TERT expression.

Response 1: The question mark has been removed.

Point 2. Overall, the manuscript contains many repetitions and several inaccuracies. Sections 1-4: these sections are practically introductory to the central topic of the review. They contain a superficial summary of TERT characteristics. Repetitions should be eliminated, and the text should be more focused on the main topic. 

Response 2: Repetition has been eliminated in lines 69-70, 79-80, and 111-112 and reorganised (lines 90-91, 103-104, 141-146, and 147-154) to improve focus.   Sections 1 and 2 have been reorganized and merged into a single section (pages 2-4). In addition, lines 216, 229-230, 245-246, 274-275, 379-381, 484-486, 516, 564-566 have been removed from the manuscript. Grammatical and typographical errors have been carefully identified and corrected.

Point 3. Page 4, lines 191-194: the mutation sites -124 (C228T) and -146 (C250T) cannot be called polymorphisms. Both are mutations and should be discussed appropriately.

Response 3: Sentence amended. Page 5 Lines 220-222. `The cytosine to thymidine transitions at C228T and C250T are common mutations close to the transcription start site and GC- boxes, that play key roles in transcriptional regulation 73 74.`

Point 4.Page 4, lines 188-190: the complex MYC and MAX activates the TERT promoter (Takakura et al, 1999), not the complex MYC and MAD1.

Response 4: Sentence amended. Page 5 Lines 217-219. `Other important transcription factors in the regulation of TERT are c-Myc and Max which bind the E-box consensus sites (5′-CACGTG-3′) at positions -165 and +44 to regulate TERT gene activity 72.`

Point 5. Page 5, lines 199-202: the splice variant which  produces the biologically functional protein  contains both α and β regions.

Response 5: Sentence amended. Page 5 Lines 234-236. `Expression of splicing variants produces biologically functional protein containing both α and β regions with a reported correlation between telomerase activity and TERT +α+β mRNA levels 81.`

Point 6. Sections 5-8: the roles of methylation, acetylation, and non-coding RNAs in the epigenetic control of the TERT transcription are not clearly presented.

Response 6: Sections 5-8 have been substantially edited.

Point 7. The authors should detail the role of hypermethylation of TERT promoter resulting in the increase of TERT transcription. This aspect deserves particular attention.

Response 7: To address the points detailed above we have inserted the following text; Page 9 Lines 392-403.

“Mechanistically, hypermethylation of TERT promoter reduces the ability of CTCF (CCCTC-binding factor) transcriptional repressors from binding the CCCTC binding regions in the first exon of TERT (Fig. 2) thereby preventing CTCF driven inhibition of TERT gene expression 130. Treatment of telomerase-positive cells with 5-azacytidine enabled CTCF binding to TERT promoter and resultant down-regulation of expression 131. Consistent with above, sulforaphane, a potent histone deacetylase inhibitor, that also down-regulates DNMTs, drove CpG demethylation and hyperacetylation of the regulatory region (from TSS, −202 to +106) of TERT and promoted binding of Mad1 and CTCF to the TERT promoter in MCF-7 and MDA-MB-231 human breast cancer cells 132. Similarly, the WT1 repressor downregulated TERT transcription in renal carcinoma cell line where overexpression of WT1 suppressed both TERT and MYC mRNA levels via binding to their promoters 133. In summary, hypermethylation of specific GpG regions of the TERT promoter prevents the binding of transcriptional repressors WT1 and CTCF.`

Point 8. Page 11, lines 379-380: the sentence is not in accordance with the paper of Yu et al., 2018: TERT cooperates with the transcription factor SP1 to stimulate DNMT3B transcription, but the TERT promoter was not investigated.

Response 8: Sentence amended. Page 12 Line 433-436. `TERT synergized with the transcription factor Sp1 to stimulate the DNMT3 transcription and promoter activity in hepatocellular carcinoma while the direct impact on TERT promoter remains undetermined.`

Point 9. The role of histone acetylation on TERT transcription is really confused and contradictory; it needs to be rewritten.

Response 9: Repetitions are eliminated and section 7 reorganized. Pages 12-13 Line 466-470, 484-486 are removed.

Point 10. The conclusions should be more focused, taking into account the most important and consistent findings about the role of epigenetics in TERT expression

Response 10: The conclusion section has been rewritten to reflect the reviewer’s important observation.

Point 11. Page15 lines 502-519: the role of TERT as a target candidate for HDACi should be significantly revised and better defined

Response 11: In response to the above point we have added/amended the following text. Page 16-17. Lines. 597-621. `The current diversity of epigenetic tools available creates an opportunity for novel strategies for reversible telomerase regulation and targeted telomere maintenance pathways. For example, preclinical and clinical information has suggested that histone deacetylase inhibitor (HDACi) and RNA targeting create a new therapeutic concept for biomedical applications 212. HDACi mediated telomerase intervention is as an example of an epigenetic focused therapy. HDACi reactivated transcriptionally silenced apoptosis via both the extrinsic and intrinsic pathways, also induced tumour suppressor gene expression including p21CIP1/WAF1 and TP53 213. Increased expression of HDAC1, HDAC2, HDAC3 and HDAC6 are reported in several cancers including colon, prostate and breast cancer and inhibition of HDACs potentially could target proliferation, differentiation, angiogenesis, and migration 214. Mechanistically the inhibition of HDACs increases acetylation of histone and non‐histone proteins, leading to an increase in transcriptionally active, open chromatin. The Food and Drug Administration (FDA) has approved several HDAC inhibitors as anti-cancer drugs including Vorinostat 215, Romidepsin 216217, Belinostat 218, and Panobinostat 219. One attractive candidate for HDACi-mediated inhibition is telomerase. Vorinostat downregulated TERT and repressed telomerase activity in non-small cell lung cancer and lymphoma cells 220,221. HDACi inhibited telomerase activity and downregulated TERT in childhood brain tumour cells 222, human normal TERT-immortalised fibroblasts, brain cancer cell lines 170, small-cell lung cancer lines 223 and prostate cancer cells 224. Conflictingly, HDAC inhibitor trichostatin A treatment also induced significant expression of TERT mRNA and telomerase activity in somatic cells where overexpression of Sp1 enhance responsiveness. However, the same treatment regime did not affect telomerase activity or TERT expression levels in cancer cell lines 168. Moreover, HDAC9 promoted ALT pathway by increased formation of ALT-associated promyelocytic leukemia nuclear bodies in ALT positive cells 225. Simultaneous targeting of ALT and telomerase positive cells with HDACs may present a viable and dual mechanism for cancer therapy.`

Point 12. All genes need to be indicated following the HUGO gene nomenclature guidelines (for example TERT and not hTERT, MYC and not c-Myc).

Response 12: All gene nomenclature has been revised within the manuscript.

Point 13. The acronym for TRF2 on page 1, line 42 needs to be provided the first time the gene is named.

Response 13: Page 2. line 73. TRF2 (Telomeric Repeat-binding factors 2)

Point 14. On page 8, line 341, UTSS stands for upstream transcription start site

The manuscript contains several grammar errors and typos.

Response 14: Page 8, line 384. The upstream transcription start site. Grammatical and typographical errors have been carefully identified and corrected

Reviewer 3 Report

The authors should be commended for their in-depth review covering the major breakthroughs of epigenetic telomerase regulation. My only major criticism is that the review could be more succinct. There is a lot of space devoted to explaining core concepts of histone modifications (etc.) which the target audience of the review should already be aware of. Referencing previous reviews on the subject should be enough to bring them up to speed if they need a reminder.

Minor comments

  • lines 77-78: It is not always the case that stem cells have longer telomere lengths than cancer cell lines.You have ALT cells and some really long telomerase positive cell lines. Please make the statement more general.
  • lines 92-97: This section is a little confusing to read. You mention the two main subunits of telomerase, then make mention of telomerase associated proteins without introducing them.
  • lines 125-127: I'm not sure of the importance of TERRA in regulating human telomerase activity. Could the authors please specify if this is in mouse?

Author Response

We thank the panel of reviewers for their constructive feedback. We have detailed in the attachment the manner, point by point, in which these have been addressed. Please see the attachment and also revised version of the manuscript.

Point 1. My only major criticism is that the review could be more succinct. There is a lot of space devoted to explaining core concepts of histone modifications (etc.) which the target audience of the review should already be aware of. Referencing previous reviews on the subject should be enough to bring them up to speed if they need a reminder.

Response 1: A brief description of concepts of histone modification has been included (Page 7. Lines 305-306; Page 12. Lines 464-466) and previous reviews have been cited to aid comprehension across all levels of readers; learner to specialist. Repetition has been eliminated to make the review more succinct.

Page 7. Lines 305-306. `Histone modification is described by covalent post-translational modification of histones including methylation, phosphorylation, acetylation, ubiquitylation, and sumoylation.`

Page 12. Lines 464-466. `DNA is packaged around histone proteins and modifications of  histone tails regulate chromatin structure and gene accessibility, therefore, modifications within the histone core affect the interactions between the nucleosome and the DNA 154.`

Point 2. lines 77-78: It is not always the case that stem cells have longer telomere lengths than cancer cell lines.You have ALT cells and some really long telomerase positive cell lines. Please make the statement more general.

Response 2: Page 3. Lines 118-120. ‘Telomere length is generally longer in stem cell populations than in somatic and cancer cell lines except for some ALT cells and some telomerase positive cell lines 39.

Point 3. lines 92-97: This section is a little confusing to read. You mention the two main subunits of telomerase, then make mention of telomerase associated proteins without introducing them.

Response 3: Page 2. Lines 51-59. ‘The telomerase complex consists of two main subunits and a range of associated proteins. These main subunits are the highly conserved, catalytic subunit, TERT protein (1132 amino acids, 127 kDa) 6 and the telomerase RNA component (TERC), where both components are required for telomerase activity 7. Telomerase associated and DNA repair protein Ku (a heterodimer of Ku70 and Ku80 subunits) interacts with telomerase through interaction with hTERT and TERC subunits 8,9. TERC and telomerase-associated proteins are constitutively expressed 10 indicating that enzyme activity depends on transcriptional regulation of TERT as the rate-limiting component of telomerase activity.’

Point 4. lines 125-127: I'm not sure of the importance of TERRA in regulating human telomerase activity. Could the authors please specify if this is in mouse?

Response 4: Page 15 lines 551-555.

‘TERRA, transcribed by RNA polymerase II from telomeres, localizes at chromosome ends and plays a role in the regulation of telomerase activity and heterochromatin formation at mammalian chromosome ends in both human and mouse models 188 189.

Reviewer 4 Report

Understanding the mechanisms of telomerase regulation is a fundamental aspect in normal and cancer cell telomere biology. The mechanisms of telomerase regulation are particularly complexes, being the expression and function of TERT (the catalytic component of telomerase and the rate-limiting factor of telomerase activity) regulated at multiple levels, including TERT promoter activity, post-transcriptional and post-translational modifications. In the present review paper, the authors have tentatively summarized the role of epigenetics in the regulation of telomerase expression and activity.

Sections 1 and 2 are poorly organized. For instance, text at lines 52-69 should be moved after lines 90-103 and both moved after the introductory part of section 2. Anyway, section 1 and 2 should be merged in a single section with all the redundancies removed.

Section 3 appears pretty out of the context, while the text related to the splicing variants (line 198-209) is very rudimentary and confounding. Basic concepts on what are TERT splicing variants and their role in regulating telomerase activity must be better presented. But, most important, the link between the alternative splicing and the epigenetic control of TERT expression should be clearly presented.

Section 5 and 6 should be merged in one single section. In addition, the interplay between Myc-dependent transcriptional activation and the CTCF/WT1-dependent transcriptional repression of TERT transcription in the presence of hypermethylated TERT promoter should be better presented (see lines 344-349).

In Section 7, the role of thrimethylated H3K9 (an epigenetic silencing marks, founds also in senescent cells) in increased gene expression should be better clarified (see lines 424-425).

All the aspect related to non-coding RNAs (including long non-coding RNAs and miRNAs) have been superficially reported throughout the main text. In particular, section 8 appears rather incomplete and the role of miRNA regulating TERT merely listed in Table 3. Have TERT been experimentally validated as a direct target of all these miRNAs? Are there miRNAs that impinge on TERT expression/function through the regulation of epigenetic factors?

Section 9 is scanty. Have FDA-approved HDACi been reported to impact on TERT expression in any study? Tricostatin A appears as an unsuitable example of HDACi to be used to target TERT. The concepts reported at lines 531-535 sound a bit out-of-date: TERT exerts also non-telomeric functions. Other than a possible application of dCas9/Tet1 combination, are there additional, epigenetic-based approaches to exploit TERT promoter as a target in cancer therapy (e.g., G-quadruplexes).

List of references should be adjusted: numbering or alphabetical order?

Lines 23-25: please adjust the concepts.

Lines 104-106 should be removed.

Line 183: please explain what “dual role feature” stands for.

Lines 191-194: It is correct to define C228T and C250T point mutations as Single nucleotide polymorphisms?

Lines 234-235 and lines 322-323 non-sense sentences.

Lines 338-340: this methodological notions is not necessary.

Overall, the way this topic has been faced in the present paper appears rather superficial. The manuscript result pretty unfocused and redundant. Take home messages are pretty unclear.

Author Response

We thank the panel of reviewers for their constructive feedback. We have detailed in the attachment the manner, point by point, in which these have been addressed. Please see the attachment and also revised version of the manuscript.

Reviewer 5 Report

The review article by Dogan and Forsyth describes regulation of hTERT gene expression by transcription factors and epigenetic mechanisms: DNA methylation, histone post-translational modifications and micro-RNAs. I have at least 4 major critical comments concerning various aspects of the manuscript.

  1. The term/terminology issue. Many terms in the manuscript are used inadequately. Examples are given below:

- The terms TERT and hTERT(human TERT) are used randomly. Authors should first define these two terms and use them consistently.

- It is sometimes hard to guess when the Authors mean (h)TERT the gene and when (h)TERT the enzyme. Gene name should be given in italic or, alternatively, described each time as “ the (h)TERT gene”; the term „promoter” should be used in conjuction with „gene” i.e. „gene promoter”; therefore instead of “hTERT promoter” this should be “hTERT gene promoter” or “hTERT promoter”.

- Line 109 and elsewhere: the term „overexpressed” should be rather limited to describe artificially (transfection/transduction etc) induced expression.

- Line 178-179: histone designation should be consistent with line 174.

- Line 192: „−124 bp ( C228T) and −146 bp (C250T)”, why are mutations in gene promoter indicated by a code designated for amino acids?

- Various histone designations are used e.g. H3K4, H3-K4 and even K4-H3.

- Line 332 „500 bases upstream of TSS into the first exon”? Impossible to locate the region. Also (line 185) it is hard to locate the region „ 1877 and -1240 upstream of the promoter region” since all +/- positions in the gene promoter should refer to TSS (Transcription start Site).

  • Line 438: TSS is given as (-150 to +150).

  1. Text clarity:

  • The first part of the manuscript is sort of chaotic, pieces of information pertaining to the same issue are dispersed in the text.

  • Line 12: the phenomena/processes influenced by telomerase should be categorized for example: cell ? organismal? survival, cell apoptosis, cell differentiation, organismal development etc.

  • Line 31: the abbreviation hTERT/TERT should be introduced here.

  • Lines 90-96: The TERC component should be introduced and defined earlier.Line 67: a very long sentence, commas needed; it is also awkward to call hESCs „caveats”.

  • A more rigorous punctuation would increase text readability.

3. Phrases/wording: There are some rather awkward phrases/wording, or simply lapsuses/errors in the text eg.:

- Line 13: „reverse transcriptase enzyme(hTERT) encodes”;

- Line 153: „TERT promoter expression”

- Line 187: „through precise mechanism that remains unclear”; if unclear, can it be precise?

- Line 48: „protects from sites of DNA damage”?

-line 397: „deactivation of H3K4 dimethylation”

4. Factual errors. The main weakness of the manuscript is the presence of (too) many factual errors, which reduce its quality. Some of them are listed below:

-Line 31: a ribonuclein is RNA- not a DNA-protein complex

- Line 218: „histone modifications'” are listed alongside „post-translational histone modifications”. What is the difference?

- Line 241: „addition of methyl or hydroxymethyl groups”. Hydroxymethyl groups are not added to DNA (cytosines) but arise due to stepwise oxidation of methyl groups.

- Line 406: the Authors seem not to distinguish between „CpG” and „CpG island” since on several instances they write that there are „CpG islands” in the TERT gene promoter while, according to Dessain et al. 2000, Zinn et al. 2007 and others, there is clearly one CpG island (with numerous CpGs).

- Line 339: „unmethylated cytosines are converted into uracils (and subsequently to thymines)” . Uracil is not converted to thymine by reaction with sodium bisulfite.

- Line 424: „trimethylation of H3K9 indicates increased gene expression”. H3K9me3 is a repressive histone modification; the Authors state so several times in Table 2.

Author Response

(The authors gave the same response as above.)
